# Robust Semi-Supervised Learning when Not All Classes have Labels

**Lan-Zhe Guo**[*], **Yi-Ge Zhang**[*], **Zhi-Fan Wu**, **Jie-Jing Shao**, **Yu-Feng Li**[†]
National Key Laboratory for Novel Software Technology, Nanjing University, Nanjing, China
{guolz,zhangyg,wuzf,shaojj,liyf}@lamda.nju.edu.cn

## Abstract

Semi-supervised learning (SSL) provides a powerful framework for leveraging unlabeled data. Existing SSL typically requires all classes have labels. However, in many real-world applications, there may exist some classes that are difficult to label or newly occurred classes that cannot be labeled in time, resulting in there are unseen classes in unlabeled data. Unseen classes will be misclassified as seen classes, causing poor classification performance. The performance of seen classes is also harmed by the existence of unseen classes. This limits the practical and wider application of SSL. To address this problem, this paper proposes a new SSL approach that can classify not only seen classes but also unseen classes. Our approach consists of two modules: *unseen class classification* and *learning pace synchronization*. Specifically, we first enable the SSL methods to classify unseen classes by exploiting pairwise similarity between examples and then synchronize the learning pace between seen and unseen classes by proposing an adaptive threshold with distribution alignment. Extensive empirical results show our approach achieves significant performance improvement in both seen and unseen classes compared with previous studies.

## 1 Introduction

Machine learning, especially deep learning, has achieved great success in various tasks by leveraging sufficient labeled training data [21]. However, for many practical tasks, it can be difficult to attain a number of labeled examples due to the high cost of the data labeling process [41, 23], which limits the widespread adoption of machine learning techniques.

Semi-supervised learning (SSL) [43] provides a powerful framework for leveraging unlabeled data when labels are limited or expensive to obtain. There has been a rapid development of SSL methods in recent years, such as entropy minimization methods [22, 8], consistency regularization methods [24, 29, 20, 32], and holistic methods[30, 2, 1, 35, 37]. It has been reported that in certain cases, such as image classification [30], SSL methods can achieve the performance of purely supervised learning even when a substantial portion of the labels in a given dataset have been discarded.

All the positive results of SSL, however, are based on a basic assumption that there are labels for each of the classes that one wishes to learn, i.e., all training and testing data are from seen classes that are observed in the labeled dataset. However, in many real-world applications, particularly those involving open-environment scenarios [42, 11], such an assumption is difficult to hold. For example, in the product recognition task, thousands of new types of products are introduced to the supermarkets every once in a while, and it would be expensive to label them all in time; in the judicial sentencing task, some sentencing elements are naturally scarce, resulting in labeled judgment documents being difficult to obtain for these elements in the training phase.

---

[*]Contribute to this work equally
[†]Corresponding author

36th Conference on Neural Information Processing Systems (NeurIPS 2022).

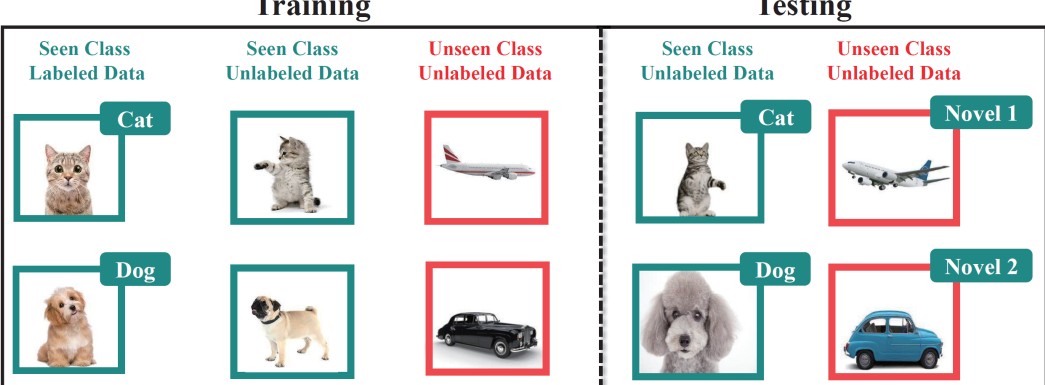

Figure 1: SSL when not all classes have labels. Training data includes labeled examples from a few seen classes as well as unlabeled examples from both seen and unseen classes. Testing data includes unlabeled examples from both seen and unseen classes. The goal is to classify not only seen class examples into accurate classes but also partition unseen class examples into proper clusters.

It is evident that existing SSL methods will misclassify unlabeled examples from unseen classes into seen classes. Even on the seen classes, SSL performance can also degrade severely with the presence of unseen classes unlabeled data [25, 10, 40]. The problem is related to open-set SSL and novel class discovery (NCD) studies. Open-set SSL [10, 6, 36, 28] aims to decrease the negative impact of unseen classes and maintain the performance robustness of seen classes. However, these methods simply detect and drop examples from unseen classes and fail to classify them. NCD [13, 15, 38] aims to discover unseen classes automatically. However, they ignore the classification task on seen classes, which results in performance degradation in seen classes.

An illustration of the problem concerned in this paper is presented in Figure 1. It is evident that both existing SSL methods and NCD methods could not tackle the problem. This inspires us to consider answering the following question in this study:

**Can we design an robust SSL algorithm that can classify both seen and unseen classes when not all classes have labels in the training data?**

To this end, we propose a new SSL method called NACH, which consists of two key modules: unseen class classification and learning pace synchronization. Specifically, we first propose a novel unseen class classification loss that can exploit pairwise similarity to classify similar example pairs into the same class and eliminate noisy pairs based on a similarity filter. We then adopt an adaptive threshold with distribution alignment to alleviate the issue that different learning paces between seen and unseen classes. Experimental results on CIFAR-10, CIFAR-100, and ImageNet-100 datasets show that NACH achieves 37.7% improvement in unseen classes compared with SSL methods, and 26.3% improvement in seen classes compared with NCD methods.

## 2 Related Work

**Semi-Supervised Learning.** SSL assumes all training and testing data are from seen classes, no matter whether labeled or unlabeled, and the goal is to classify unlabeled examples into the ground-truth classes. SSL has a long research history [4]. Our paper is mainly related to deep SSL, which introduces SSL techniques to deep neural networks and has achieved significant advancement in recent years. The mainstream of deep SSL can be broadly categorized into entropy minimization methods [22, 8], consistency regularization methods [24, 29, 20, 32], and holistic methods [30, 2, 1]. When not all classes have labels in the training data, these methods will misclassify unlabeled data from unseen classes as the seen classes and fail to address the problem concerned in this paper.

**Open-Set Semi-Supervised Learning.** Open-set SSL relaxes the assumption of SSL and considers a more practical scenario that training data could contain unseen class unlabeled examples. However, they still assume all testing examples are from seen classes, and the goal is to decrease the negative impact of unseen class unlabeled data in order to maintain the performance robustness in seen

classes. Many open set SSL methods have been proposed in recent years [10, 6, 36, 31, 28, 16, 26, 17], such as DS3L [10], which assign weights to unlabeled data based on a bi-level optimization, UASD [6], which filter unlabeled examples based on the prediction consistency, MTC [36], which adopt a multi-task curriculum learning framework to detect unseen classes and classify seen classes simultaneously, T2T [16], which propose a novel cross-modal matching strategy to detect unseen classes. OpenMatch [28], which unify FixMatch algorithm with novelty detection based on one-vs-all (OVA) classifiers. However, these methods can still not classify unseen classes.

**Novel Class Discovery.** NCD assumes training data consists of seen class labeled examples and unseen class unlabeled examples, and the goal is to classify both seen and unseen classes during the testing phase. The NCD problem is first formally introduced in [13]. Recently, many NCD methods have been proposed based on a two-step training strategy [13–15, 7, 12, 39, 38], i.e., a data embedding is learned on the labeled data using a metric learning technique, and then fine-tuned while learning the cluster assignments on the unlabeled data. In contrast to the problem studied in this paper, NCD methods ignore the abundant seen unlabeled examples that are usually easy to collect in real-world applications.

## 3 Preliminary and Background

Give the training data which contains $n$ labeled examples $\mathcal{D}_l = \{(\mathbf{x}_1, \mathbf{y}_1), \cdots, (\mathbf{x}_n, \mathbf{y}_n)\}$ and $m$ unlabeled examples $\mathcal{D}_u = \{\mathbf{x}_{n+1}, \cdots, \mathbf{x}_{n+m}\}$. Usually, $m \gg n$. $\mathbf{x} \in \mathbb{R}^D, \mathbf{y} \in \mathcal{Y} = \{1, \cdots, C_L\}$ where $D$ is the feature dimension and $C_L$ is the number of seen classes. We use $C_U$ to represent the total number of classes in unlabeled data, Previous SSL studies assume $C_L = C_U$ and NCD assumes $C_L \cap C_U = \emptyset$. In this paper, the number of seen classes $C_{seen} = C_L \cap C_U$ and the number of unseen classes $C_{unseen} = C_U \setminus C_L$. The goal is to learn a classification model $f(\mathbf{x}; \theta)$ from training data. Specifically, the $f(\mathbf{x}; \theta)$ can be decomposed of a representation learning model $g(\mathbf{x}; \theta) : \mathbb{R}^D \to \mathbb{R}^d$ to learn a low-dimensional feature $z$ and a classification model $h(\mathbf{z}) : \mathbb{R}^d \to \mathbb{R}^{C_{seen}+C_{unseen}}$.

The training loss of an SSL algorithm usually contains supervised loss $\mathcal{L}_s$ and unsupervised loss $\mathcal{L}_u$ with a trade-off parameter $\lambda_u > 0$: $\mathcal{L}_s + \lambda_u \mathcal{L}_u$, where $\mathcal{L}_s$ is constructed on $\mathcal{D}_l$ and $\mathcal{L}_u$ is constructed on $\mathcal{D}_u$. Typically, $\mathcal{L}_s$ applies the standard cross-entropy loss on labeled examples:

$$\mathcal{L}_s = \frac{1}{n} \sum_{i=1}^{n} H(\mathbf{y}_i, p(\mathbf{x}_i)) \tag{1}$$

where $p(\mathbf{x}) = \text{Softmax}(f(\mathbf{x}; \theta))$ indicate the predicted probabilities produced by the model $f$ for the input $\mathbf{x}$, and $H(\cdot, \cdot)$ is the cross-entropy function.

Different constructions of the unsupervised loss $\mathcal{L}_u$ lead to different SSL algorithms. Typically, there are two ways of constructing $\mathcal{L}_u$: one is to assign pseudo-labels to formulate a "supervised loss" such as the cross-entropy loss, and another one is to optimize a regularization that does not depend on labels such as consistency regularization.

Take the FixMatch [30] and UDA [34] for examples, FixMatch adopts the pseudo-label loss which can be written as:

$$\mathcal{L}_u = \frac{1}{m} \sum_{i=n+1}^{n+m} \mathbb{I}\left(\max\left(p(\alpha(\mathbf{x}_i))\right) \geq \tau\right) H\left(\widehat{\mathbf{y}}_i, p(\mathcal{A}(\mathbf{x}_i))\right) \tag{2}$$

where $\alpha(\mathbf{x})$ and $\mathcal{A}(\mathbf{x})$ indicate the weak and strong augmentation, $\widehat{\mathbf{y}}_i = \arg\max p(\alpha(\mathbf{x}_i))$ represent the pseudo-label for unlabeled example $\mathbf{x}_i$, $\tau$ is the confidence threshold for pseudo-label selection, $\mathbb{I}(\cdot)$ is the indicator function.

UDA [34] adopts the consistency regularization which can be written as

$$\mathcal{L}_u = \frac{1}{m} \sum_{i=n+1}^{n+m} \|p(\text{aug}(\mathbf{x}_i)) - p(\text{aug}'(\mathbf{x}_i))\|_2^2 \tag{3}$$

where $\text{aug}(\cdot)$ and $\text{aug}'(\cdot)$ represents different augmentation strategies.

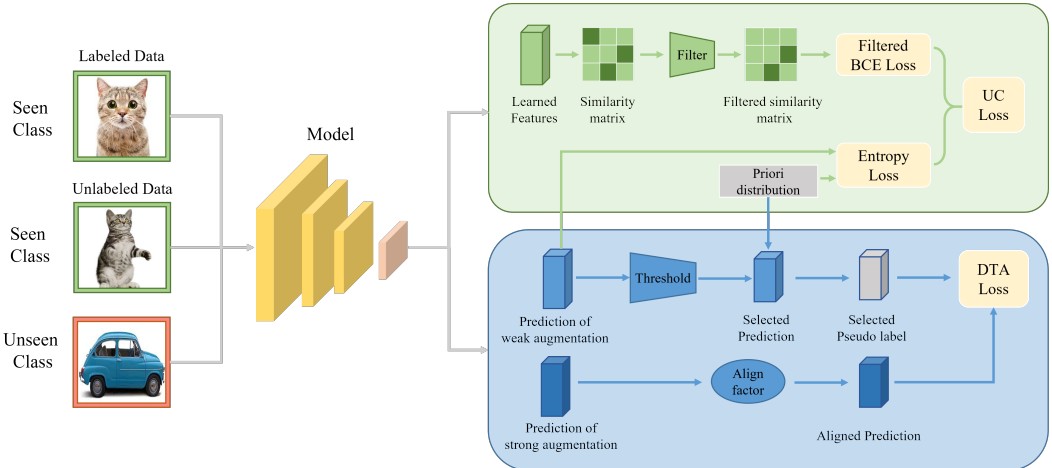

Figure 2: Framework of the proposed NACH algorithm. The unsupervised loss of NACH can be decomposed into $\mathcal{L}_{UC}$ and $\mathcal{L}_{DTA}$. The flowchart of how to compute $\mathcal{L}_{UC}$ and $\mathcal{L}_{DTA}$ lies in the green box (top) and blue box (bottom), respectively.

# 4 The NACH Method

In this section, we present the proposed NACH method. The overall framework of NACH is illustrated in Figure 2. Specifically, NACH consists of two main parts: i) unseen class classification, which contains a new unsupervised loss to discover unseen classes automatically; ii) learning pace synchronization, which contains an adaptive threshold with distribution alignment to synchronize the different learning paces between seen and unseen classes. We first provide an overview of the learning objective. The concrete details of the objective are provided in the following.

## 4.1 NACH: An Overview for Two Problems

Previous SSL methods do not have the ability to classify unseen classes, which leads to a number of examples from unseen classes being misclassified into seen classes when not all classes have labels. To address this problem, two main challenges need to be considered.

The first one is how to automatically classify unseen classes during model training. We propose to cluster unlabeled examples using the pairwise objective [12], and then we adjust the results of the clustering using a known prior distribution. Specifically, we adopt the cosine similarity to find the most similar example in a mini-batch for each example as the positive pairs. To avoid the mismatched situation where an example from seen classes and an example from unseen classes are wrongly paired, we design a similarity-based filter to get rid of the appearance of such noisy pairs.

The second is how to synchronize the different learning paces caused by the different learning styles between seen and unseen classes. We propose a metric to measure the difference between the learning status of seen and unseen classes. Then, we formulate this metric into an adaptive threshold with distribution alignment. By selecting pseudo-labels with the adaptive threshold, the model can adjust the predicted probability adaptively to synchronize the learning differences between seen classes and unseen classes.

Moreover, we adopt the contrastive learning method SimCLR [5] to pre-train the backbone network on the whole dataset in an unsupervised fashion in order to learn a good representation.

Overall, the objective of NACH consists of three parts: i) a supervised loss $\mathcal{L}_s$ for labeled data; ii) an unseen classes classification loss $\mathcal{L}_{UC}$ by exploiting pairwise similarity to classify unseen classes; iii) an unsupervised loss $\mathcal{L}_{DTA}$ by assigning pseudo-labels using the adaptive threshold to achieve robust performance on both seen and unseen classes:

$$\mathcal{L} = \mathcal{L}_s + \lambda_1 \mathcal{L}_{UC} + \lambda_2 \mathcal{L}_{DTA} \tag{4}$$

where $\lambda_1$ and $\lambda_2$ are trade-off hyper-parameters, which are all set to 1 in default.

## 4.2 Unseen Class Classification

To enable SSL the ability to classify unseen classes, one way is to adopt the binary cross-entropy (BCE) loss which can effectively exploit the pairwise similarity. The classical BCE loss can be written as:

$$\mathcal{L}_{BCE} = -\frac{1}{(m+n)^2} \sum_{i=1}^{m+n} \sum_{j=1}^{m+n} \left[ s_{ij} \log p(\mathbf{x}_i)^\top p(\mathbf{x}_j) \right.$$
$$\left. + (1 - s_{ij}) \log \left( 1 - p(\mathbf{x}_i)^\top p(\mathbf{x}_j) \right) \right] \tag{5}$$

where $s_{ij}$ is a measure of the similarity between $\mathbf{x}_i$ and $\mathbf{x}_j$ (e.g., cosine similarity). The first term aims to pull two similar examples closer, while the latter term aims to push two dissimilar examples farther apart. Taking advantage of the BCE loss, we can achieve clustering of unseen classes by pairing similar examples.

This BCE loss is commonly adopted in NCD studies. However, in our study, directly using the BCE loss can not effectively classify unseen classes since both seen and unseen classes are in the unlabeled data. Due to the different learning paces between seen and unseen classes, two examples belonging to the same unseen class are likely to be pushed apart at the early training stage due to their low similarity. To address this issue, we neglect the push-apart strategy and adopt the cosine similarity to find the most similar example to be pulled together for each example in a batch [3]. Moreover, to further improve the pair accuracy, we propose a new filter strategy to decrease the wrong pairs that consist of examples from seen and unseen classes.

Specifically, given a batch of $B$ labeled examples $\{(\mathbf{x}_b^l, \mathbf{y}_b^l) : b \in (1, \cdots, B)\}$ and a batch of $\mu B$ unlabeled examples $\{\mathbf{x}_b^u : b \in (1, \cdots, \mu B)\}$ where $\mu$ determines the relative batch size of labeled and unlabeled data. For labeled examples, we can match pairs based on their ground-truth labels. For unlabeled example $\mathbf{x}_b^u$, we first find the example $\widetilde{\mathbf{x}}_b^u$ which is the most similar to $\mathbf{x}_b^u$ according to their representation, then compute the cosine similarity between $\widetilde{\mathbf{x}}_b^u$ and the batch of labeled examples: $\{\cos(g(\widetilde{\mathbf{x}}_b^u), g(\mathbf{x}_1^l)), \cos(g(\widetilde{\mathbf{x}}_b^u), g(\mathbf{x}_2^l)), \cdots, \cos(g(\widetilde{\mathbf{x}}_b^u), g(\mathbf{x}_B^l))\}$. We sort the similarity in descending order and obtain $\{d_1, d_2, \cdots, d_B\}$, where $d_i$ indicates the $i$-largest similarity. Consider the fact that if $\mathbf{x}^u$ is an example from unseen classes and $\widetilde{\mathbf{x}}^u$ is an example from seen classes, then $\cos(g(\mathbf{x}^u), g(\widetilde{\mathbf{x}}^u))$ is likely to be less than some values $d_k, k \in \{1, \cdots, B\}$. Based on this phenomenon, we can set a threshold $k$, and retain the example $\mathbf{x}_b^u$ only when two examples $\mathbf{x}_b^u$ and $\widetilde{\mathbf{x}}_b^u$ satisfy $\cos(g(\mathbf{x}_b^u), g(\widetilde{\mathbf{x}}_b^u)) \geq d_k$. Then the filtered BCE loss ($\mathcal{L}_{FBCE}$) can be obtained as follows:

$$\mathcal{L}_{FBCE} = -\frac{1}{B} \sum_{b=1}^{B} \log \left( p(\mathbf{x}_b^l)^\top p(\widetilde{\mathbf{x}}_b^l) \right)$$
$$- \frac{1}{\mu B} \sum_{b=1}^{\mu B} \mathbb{I}(\cos(g(\mathbf{x}_b^u), g(\widetilde{\mathbf{x}}_b^u)) \geq d_k) \log \left( p(\mathbf{x}_b^u)^\top p(\widetilde{\mathbf{x}}_b^u) \right) \tag{6}$$

Moreover, inspired by [3], we regularize the predictive distribution of the pseudo-label to be close to a prior probability distribution $\mathcal{P}$ of labels $\mathbf{y}$ to prevent the model from classifying all unseen classes into one class and thus hindering the performance of unseen classes classification:

$$\mathcal{L}_{ENT} = \mathrm{KL}\left( \frac{1}{B} \sum_{b=1}^{B} p(\mathbf{x}_i^l) + \frac{1}{\mu B} \sum_{b=1}^{\mu B} p(\mathbf{x}_b^u) \big\| \mathcal{P}(\mathbf{y}) \right) \tag{7}$$

The definition of $\mathcal{L}_{UC}$ can be written as:

$$\mathcal{L}_{UC} = \mathcal{L}_{FBCE} + \mathcal{L}_{ENT} \tag{8}$$

## 4.3 Learning Paces Synchronization

For the learning of seen classes, the supervision is from the ground-truth labels. However, for unseen classes, the model can only learn from the pairwise objective. This results in the learning pace of unseen classes being slower than seen classes. To maintain robust performance in seen classes, we propose to synchronize the learning paces between seen and unseen classes. Previous

SSL algorithms, e.g., FixMatch [30], select pseudo-labels based on a fixed confidence threshold. In this paper, we propose an adaptive threshold to synchronize the learning paces adaptively. Different from previous SSL methods with adaptive threshold [35, 37, 9], we assign different threshold for seen and unseen classes separately in order to synchronize their learning paces. First, we define a metric to measure the difference in learning status between seen and unseen classes. We calculate the maximum classification confidence and the corresponding pseudo-label for each example:

$$\widehat{p}_i = \max(p(\alpha(\mathbf{x}_i))), \qquad \widehat{\mathbf{y}}_i = \arg\max(p(\alpha(\mathbf{x}_i))) \tag{9}$$

Then we define the metric $\mathbf{U}$ as follows:

$$\mathbf{U} = \Big(\frac{1}{N_{seen}} \sum_{\mathbf{x}_i \in \mathcal{X}_{seen}} \widehat{p}_i\Big) - \Big(\frac{1}{N_{unseen}} \sum_{\mathbf{x}_j \in \mathcal{X}_{unseen}} \widehat{p}_j\Big) \tag{10}$$

where $N_{seen}$ and $N_{unseen}$ denote the total number of examples which are classified as seen classes and unseen classes. $\mathcal{X}_{seen}$ refers to examples with pseudo-label belongs seen classes and $\mathcal{X}_{unseen}$ refers to examples with pseudo-label belongs to unseen classes.

$\mathbf{U}$ can effectively assess the degree of learning difference between seen classes and unseen classes. We applied $\mathbf{U}$ to adaptively adjust the confidence threshold for unseen classes. For seen classes we assume that the confidence threshold is $\tau$, then for unseen classes, we heuristically set the threshold to $\tau - \beta\mathbf{U}$ where $\beta$ is the trade-off parameters.

The above operation ensures that more examples predicted as unseen classes can be selected in the model training process. To further exploit the pseudo-label, we adopted an distribution alignment strategy. It should be noted that although $\mathcal{L}_{ENT}$ already takes into account distribution alignment, that is for all examples. Here we consider distribution alignment based on examples with confidence above the threshold.

Our goal is to have the distribution of these above-threshold examples converge to a known prior distribution to better learn the unseen classes, so we add distribution alignment as a fine-tuning for logits to $\mathcal{L}_{DTA}$. Specifically, we define:

$$\begin{aligned} \mathcal{P}(X_{select}) = \sum_{\mathbf{x}_i \in \mathcal{X}_{seen}} \mathbb{I}\left(\widehat{p}_i \geq \tau\right) p(\alpha(\mathbf{x}_i)) \\ + \sum_{\mathbf{x}_j \in \mathcal{X}_{unseen}} \mathbb{I}\left(\widehat{p}_j \geq \tau - \beta\mathbf{U}\right) p(\alpha(\mathbf{x}_j)) \end{aligned} \tag{11}$$

We then require $\mathcal{P}(X_{select})$ and prior distribution $\mathcal{P}(y)$ to be aligned:

$$F_{ali} = \log \mathcal{P}(X_{select})/\mathcal{P}(y) \tag{12}$$

The adjustment factor $F_{ali}$ aims to align the distribution of selected data to a prior distribution. We then compute the cross-entropy between prediction on strong augmented examples and the corresponding pseudo-label as the $\mathcal{L}_{DTA}$ loss:

$$\mathcal{L}_{DTA} = \sum_{\mathbf{x}_i \in \mathcal{X}_{seen} \cup \mathcal{X}_{unseen}} \mathbb{I}\left(\widehat{p}_i \geq \tau_i\right) H\left(\widehat{\mathbf{y}}_i, p(\mathcal{A}(\mathbf{x}_i)) + F_{ali}\right) \tag{13}$$

where $\tau_i$ is $\tau$ for $\mathbf{x}_i$ belongs to $\mathcal{X}_{seen}$ and $\tau - \beta\mathbf{U}$ for $\mathbf{x}_i$ belongs to $\mathcal{X}_{unseen}$. It is important to note that the adaptive thresholds and logit adjustment with distribution alignment factor are complementary: adaptive threshold adjustment is designed for preventing the fixed confidence threshold from hindering the learning of unseen classes. Thus, we adaptively adjust the threshold so that more examples of unseen classes could be learned by the model, which is beneficial to logit adjustment with distribution alignment. Distribution alignment as a factor for logit adjustment also makes the model less biased towards seen classes and thus facilitate the learning of unseen classes.

## 5 Experiments

In this section, we give a comprehensive evaluation of NACH. Experimental results and detailed analysis are reported to demonstrate the effectiveness of our proposal.

Table 1: Classification accuracy of compared methods on seen, unseen and all classes. The underline indicates the performance is worse than the baseline SSL methods.

| Classes | Dataset | SSL | Open-Set SSL | | NCD | | | |
| | | Fixmatch | DS3L | CGDL | DTC | RankStats | ORCA | OURS |
|---|---|---|---|---|---|---|---|---|
| **Seen** | CIFAR-10 | 71.5 | 77.6 | 72.3 | _53.9_ | 86.6 | 88.2 | **89.5** |
| | CIFAR-100 | 39.6 | 55.1 | 49.3 | _31.3_ | _36.4_ | 66.9 | **68.7** |
| | ImageNet-100 | 65.8 | 71.2 | 67.3 | _25.6_ | _47.3_ | 89.1 | **91.0** |
| | Average | 59.0 | 68.0 | 63.0 | 36.9 | 56.8 | 81.4 | **83.1** |
| **Unseen** | CIFAR-10 | 50.4 | _45.3_ | _44.6_ | _39.5_ | 81.0 | 90.4 | **92.2** |
| | CIFAR-100 | 23.5 | 23.7 | _22.5_ | _22.9_ | 28.4 | 43.0 | **47.0** |
| | ImageNet-100 | 36.7 | _32.5_ | _33.8_ | _20.8_ | _28.7_ | 72.1 | **75.5** |
| | Average | 36.9 | 33.9 | 33.6 | 27.7 | 46.0 | 68.5 | **71.6** |
| **All** | CIFAR-10 | 49.5 | _40.2_ | _39.7_ | _38.3_ | 82.9 | 89.7 | **91.3** |
| | CIFAR-100 | 20.3 | 24.0 | 23.5 | _18.3_ | 23.1 | 48.1 | **52.1** |
| | ImageNet-100 | 34.9 | _30.8_ | _31.9_ | _21.3_ | 40.3 | 77.8 | **79.6** |
| | Average | 34.9 | 31.7 | 31.7 | 26.0 | 48.8 | 71.9 | **74.3** |

## 5.1 Experimental Setup

**Datasets.** We evaluate NACH and compared methods on three SSL benchmark datasets CIFAR-10, CIFAR-100 [18] and ImageNet [27]. Specifically, for the ImageNet dataset, 100 classes are sub-sampled following [33, 3]. We first divide classes into 50% seen and 50% unseen classes, then select 50% of seen classes as the labeled data, and the rest as unlabeled data.

**Compared Methods.** We compare NACH with representative SSL, open-set SSL, and NCD methods. The SSL and open-set SSL methods are extended to be applicable to unseen classes in the following way: examples are divided into known classes and unknown classes, we report their performance on seen classes and apply K-means clustering to unseen class examples to obtain clustering results. For SSL, the FixMatch [30] is adopted due to its empirical success and estimate unseen classes based on softmax confidence scores. For open-set SSL, we adopt two representative methods DS3L [10] which tries to assign lower weights to unseen classes unlabeled data, and CGDL [31] which automatically rejects unseen class examples. NCD methods are extended to classify seen classes by using the Hungarian algorithm [19] to match some of the discovered classes with classes in the labeled data. Specifically, two NCD methods are employed: DTC [13] and RankStats [12], which have been reported to achieve the state-of-the-art performance on NCD tasks. Moreover, we also compare NACH with ORCA [3] methods, which consider a similar setting with our paper.

All compared methods are implemented based on the pre-trained model using the contrastive learning algorithm SimCLR [5]. The only exception is DTC which has its own specialized pre-training procedure on labeled data [13].

**Implementation Details.** For CIFAR datasets, we use ResNet-18 as the backbone model. The model is trained by using the standard Stochastic Gradient Descent method with a momentum of 0.9 and a weight decay of 0.0005. We trained the model for 200 epochs with a batch size of 512. Following [3], we only update the parameters of the last block of ResNet in the second training stage to avoid over-fitting. For the ImageNet dataset, we use ResNet-50 as the backbone model. The model is trained by using standard SGD with a momentum of 0.9 and a weight decay of 0.0001. We train the model for 90 epochs with a batch size of 512. For all experiments, the cosine annealing learning rate schedule is adopted. All experiments are performed on a single NVIDIA 3090 GPU.

## 5.2 Main Results

The mean classification accuracy on CIFAR-10, CIFAR-100, and ImageNet-100 dataset are provided in Table 1. From the results, it can be observed that open-set SSL methods can address the performance degradation problem on seen classes, but can not classify unseen classes accurately. NCD methods, e.g., RankStats, can improve the unseen classification performance, but suffer performance

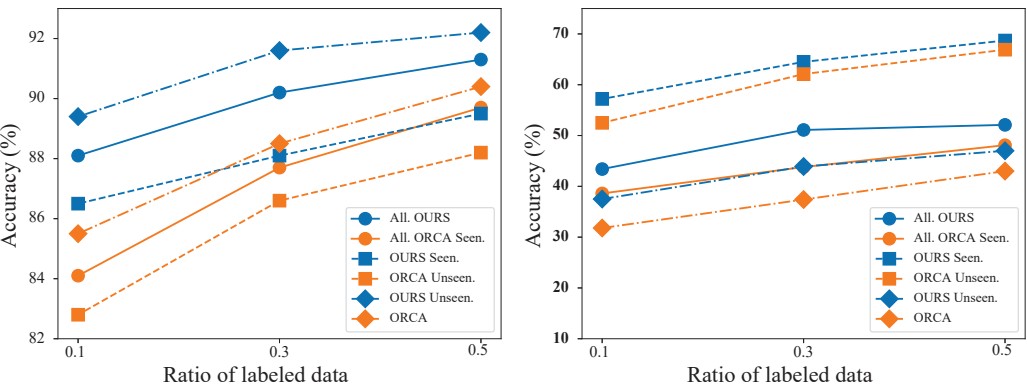

(a) Classification accuracy on CIFAR-10.  (b) Classification accuracy on CIFAR-100.

Figure 3: Performance of NACH and ORCA with different numbers of labeled data.

degradation problem on seen classes, DTC performs even worse than SSL on unseen class classification. On the contrary, our proposal NACH can not only classify unseen classes accurately but also maintain robust performance on seen classes. For example, NACH achieves a 24.1% improvement on seen classes and 34.7% on unseen classes compared with the FixMatch methods. Compared with ORCA, NACH also achieves a significant performance improvement in both seen and unseen classes.

To further demonstrate the effectiveness of our proposal with varying label sizes, we evaluate the performance of NACH and ORCA with different numbers of labeled data, as shown in Figure 3. The results show that the performance of NACH is always better than ORCA in all cases with a significant margin. Moreover, the results also demonstrate that our proposal NACH is robust with label size, for example, even with only 10% of labeled examples, the unseen-class accuracy of our method can still reach more than 89.4 and 37.5.

### 5.3 Detail Analysis

In this subsection, detailed analyzes are shown to help understand the superiority of our proposal, including analyzes of the two modules: $\mathcal{L}_{UC}$ and $\mathcal{L}_{DTA}$, and hyper-parameter sensitivity analysis.

**Analysis of Unseen-Class Classification Loss:** $\mathcal{L}_{UC}$. We first show the effectiveness of the proposed FBCE loss by proposing a basic model (BM) with $C_{seen} + C_{unseen}$ classification heads and optimize the model by directly minimizing the standard BCE loss and our proposed FBCE loss separately. The comparison results are reported in Figure 4, including the ratio of wrongly selected seen-unseen pairs and correctly selected unseen-unseen pairs during the model training process. From the results, we can see that the proposed FBCE loss can effectively decrease the ratio of wrongly selected seen-unseen pairs while selecting more correct unseen-unseen pairs. This demonstrates the effectiveness of the proposed filter strategy. We also study the effectiveness of the $\mathcal{L}_{UC}$ loss, the results are presented in Table 2. From the results, we can see the performance on both seen and unseen classes can be improved by applying the $\mathcal{L}_{UC}$ loss.

Table 2: Analysis of $\mathcal{L}_{UC}$: classification accuracy on CIFAR-100.

| Method | Seen | Unseen | All |
|---|---|---|---|
| FixMatch | 39.6 | 23.5 | 20.3 |
| BM + $\mathcal{L}_{BCE}$ | 72.8 | 28.3 | 31.5 |
| BM + $\mathcal{L}_{BCE}$ + $\mathcal{L}_{ENT}$ | 67.2 | 44.2 | 50.7 |
| BM + $\mathcal{L}_{UC}$ (UC Model) | 68.2 | 44.3 | 50.2 |

Table 3: Analysis of $\mathcal{L}_{DTA}$: classification accuracy on CIFAR-100.

| Method | Seen | Unseen | All |
|---|---|---|---|
| UC Model | 68.2 | 44.3 | 50.2 |
| UC Model + DA | 68.6 | 44.8 | 50.3 |
| UC Model + DT | 69.3 | 45.6 | 49.6 |
| UC Model + DTA | 68.7 | 47.0 | 52.1 |

**Analysis on Adaptive Threshold:** $\mathcal{L}_{DTA}$. We compare the performance between the proposed adaptive threshold and the static threshold (e.g., $\tau = 0.95$ in FixMatch), the results are reported in Figure 5. From Figure 5a, we can see that the prediction confidence between seen and unseen

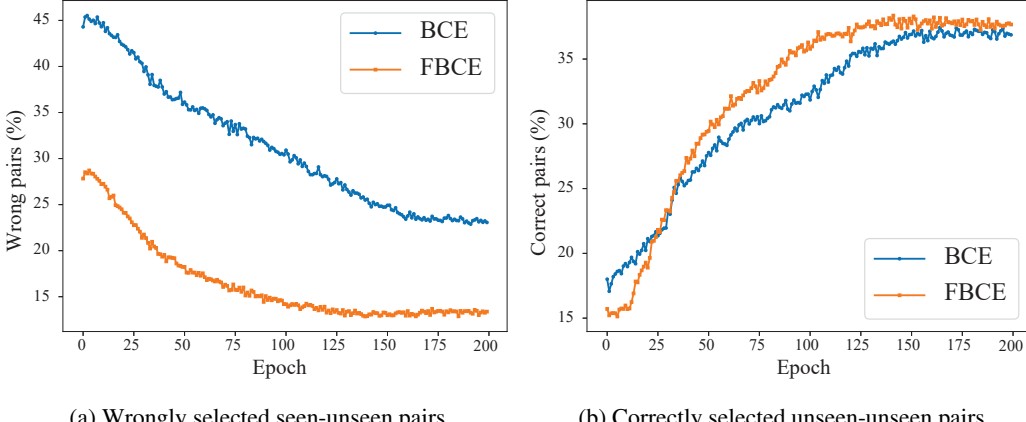

(a) Wrongly selected seen-unseen pairs      (b) Correctly selected unseen-unseen pairs

Figure 4: (a) Wrongly selected seen-unseen pairs; (b) Correctly selected unseen-unseen pairs. Both the above results are conducted on CIFAR-100 dataset.

classes are significantly different, thus, it is not proper to adopt a static threshold. Figure 5b and Figure 5c show that the proposed adaptive threshold can help select more pseudo-labels for unseen class examples and improve the accuracy of pseudo-label assignment significantly. Results in Table 3 give a more clear ablation study to demonstrate the effectiveness of our proposed $\mathcal{L}_{DTA}$ loss.

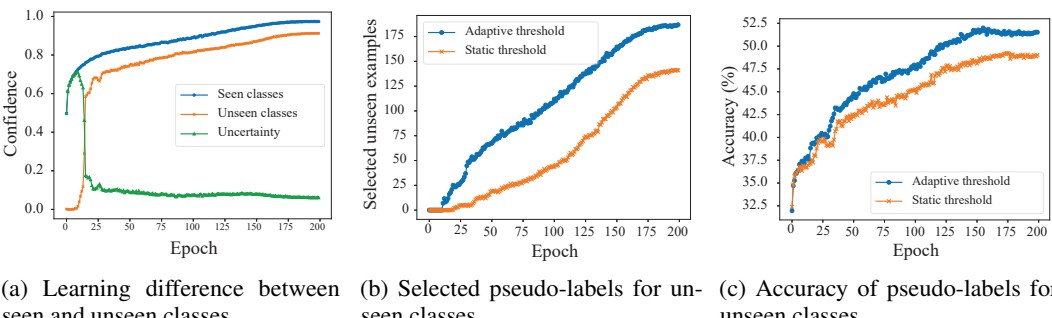

(a) Learning difference between seen and unseen classes

(b) Selected pseudo-labels for unseen classes

(c) Accuracy of pseudo-labels for unseen classes

Figure 5: (a) learning difference between seen classes and unseen classes during training; (b) number of pseudo-labels for unseen classes; (c) accuracy of pseudo-labels for unseen classes. All the above results are conducted on CIFAR-100 dataset.

## 5.4 Parameter Sensitivity Analysis

**Evaluating different $k$ used in $\mathcal{L}_{FBCE}$.** The intention of $\mathcal{L}_{FBCE}$ is to filter mismatched pairs containing examples from seen classes and unseen classes and the hyper-parameter $k$ determines the threshold of the noisy pairs filter. We provide the performance with different $k$ in Figure 6. From the results, we can see that, when $k = 2$, the proposal achieves the best performance on all classes, and the performance does not degrade severely with $k$ changes. This demonstrates that our proposal is quite robust with the selection of $k$. Moreover, we also study how the wrongly selected pairs and correctly selected pairs changes with different $k$. The results show that, the filter strategy can decrease the wrong pairs significantly compared with no filter, with the decrease of $k$ the wrong pairs are decreased. Meanwhile, the filter strategy can also increase the correct pairs. When $k = 2$ the proposal achieves a satisfying balance between the wrongly selected pairs and correctly selected pairs. so that the proposed method achieves the best performance.

**Evaluating different $\beta$ used in $\mathcal{L}_{DTA}$.** We further analyze the impact of the adaptive threshold hyper-parameter $\beta$. $\beta$ controls the margin of confidence threshold between seen and unseen classes. More pseudo-labels for unseen classes will be selected with a larger $\beta$. The results show the when $\beta = 2$ the proposal achieves the best result and the performance does not degrade severely with $\beta$ changes. This demonstrate that the proposal is quite robust to the selection of $\beta$.

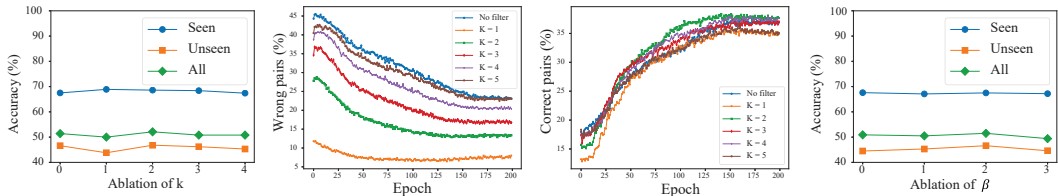

(a) Accuracy with different hyper-parameter $k$

(b) Wrongly selected seen-unseen pairs

(c) Correctly selected unseen-unseen pairs

(d) Accuracy with different hyper-parameter $\beta$

Figure 6: (a) accuracy with different $k$; (b) wrongly selected pairs with different $k$; (c) correctly selected pairs with different $k$; (d) accuracy with different $\beta$.

## 6  Conclusion

In this paper, we tackle an important and practical scenario of SSL, that is, SSL when not all classes have labels. We propose a robust SSL algorithm NACH that consists of an unseen class classification objective that can exploit pairwise similarity and eliminate noisy pairs, and an adaptive threshold with distribution alignment that can synchronize the learning paces between seen and unseen classes. Extensive experiments clearly show the effectiveness of our proposal. The code is available at `https://www.lamda.nju.edu.cn/code_NACH.ashx`

How to classify unseen classes with no labeled data is an important problem in SSL. Our work puts a promising scheme in this direction. One limitation of our scheme is it does not have theoretical guarantees. We will put efforts into this direction in future work, such as giving generalization risk analysis on unseen classes.

## Broader Impact

This paper studies the problem of SSL when not all classes have labels, which has been less investigated in SSL. We hope this work can attract more future attention to explore the robustness of SSL in more practical scenarios and promote SSL in wider applications.

## Acknowledgment

This research was supported by the National Key R&D Program of China (2022YFC3340901), the National Science Foundation of China (62176118, 61921006), and the Huawei Cooperation Fund.

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
