# OpenReview forum: "Robust Semi-Supervised Learning when Not All Classes have Labels"
_NeurIPS.cc/2022/Conference — NeurIPS 2022 Accept_

### Official Review · Reviewer_yoL8 · 2022-07-06

**Rating:** 8
**Confidence:** 4
**Soundness:** 4 excellent
**Presentation:** 3 good
**Contribution:** 3 good

**Summary:**

This paper focuses on a new semi-supervised learning problem: not all classes have labels, i.e., there are some classes that have no one labeled example. The problem is important and different from previous SSL studies. The ability of SSL to classify unseen classes is useful in real-world applications since the labels of some classes may be difficult to collect in real tasks. The learning goal is to maximize performance in unseen classes and maintain safety in seen classes. The proposed method contains an unseen class discovery loss that exploits pairwise similarity to classify unseen classes and an adaptive threshold unsupervised loss to balance the learning between seen and unseen classes. The experimental results demonstrate the effectiveness of the proposal.

**Questions:**

1.	How to relax the prior knowledge about the number of unseen classes. In real-world tasks, it may be hard to know in advance.

2.	How about the impact of $\lambda_1$ and $\lambda_2$? How to set the two parameters in real applications?


**Ethics Review Area:**

["Responsible Research Practice (e.g., IRB, documentation, research ethics)"]

**Limitations:**

The authors addressed the limitations and potential negative societal impact of their work.

**Strengths And Weaknesses:**

Strengths:
1.	This paper studies a new problem in SSL, i.e, not all classes have labeled training examples. This is a practical setting and worth studying. In my view, this paper can inspire subsequent researchers and tackle SSL in more realistic scenarios.

2.	The proposal is simple and effective. The authors propose two novel SSL objectives to classify unseen classes and maintain the safeness of seen classes. This can be simply integrated with existing SSL methods. Experimental results are convincing to demonstrate the effectiveness of the proposal.

3.	The paper is well organized and easy to follow.

Weaknesses:
1.	The proposed method needs to know the number of unseen classes. It may limit the application of the proposal.

2.	The impact of the trade-off hyper-parameters $\lambda_1$ and $\lambda_2$ need to be discussed.

---

> ### Author Response · Authors · 2022-08-02
> **Rebuttal**
>
> Thanks for your careful review, please find the response in the following.
>
> 1) The number of unseen classes
>
> The number of unseen classes in unlabeled data can be estimated via clustering methods. For example, we can cluster the unlabeled examples using k-means multiple times, varying the number of classes. And then evaluate the cluster quality to help determine the number of classes.
>
> 2) Hyperparameters
>
> Our proposal is robust to the hyperparameters. We set $\lambda_1$ and $\lambda_2$ as 1 in all experiments without specific tuning, the experimental results are promising.

---

### Official Review · Reviewer_42nR · 2022-07-07

**Rating:** 7
**Confidence:** 5
**Soundness:** 3 good
**Presentation:** 3 good
**Contribution:** 3 good

**Summary:**

This paper studies semi-supervised learning where unlabeled data contains unseen classes. Unlike previous open-set SSL methods and NCD methods, this paper aims to achieve high classification accuracy on both seen and unseen classes. This is an important problem and has not been well-studied. To address it, the authors propose a novel SSL approach called SU-SSL. The proposal exploits pairwise similarity to discover unseen classes and avoid performance degradation on seen classes via an adaptive threshold. Experimental results show that the proposal can achieve both performance improvement on seen and unseen class classification.

**Questions:**

1.	How does the performance vary with different ratios of unlabeled examples belonging to unseen classes?

**Limitations:**

The authors have provided a broader impact statement.

**Strengths And Weaknesses:**

Strengths:
1.	The problem studied in this paper is important and has not been well addressed. This paper focuses on semi-supervised learning where unlabeled data contains unseen classes. Unlike previous open-set SSL methods that only reject all unseen classes and NCD methods that only classify unseen classes, this paper aims to achieve high performance on both seen classes and unseen classes. This is an interesting problem and can promote SSL to be applied in more practical settings.
2.	The proposed method is clear and technically sound. To maximize performance on unseen classes, the proposal adopts a novel unseen class classification loss that exploits pairwise similarity to classify similar sample pairs into the same classes. To maintain safeness in seen classes, the proposal adopts an adaptive threshold with distribution alignment to alleviate the different learning paces. These two techniques can address the problem effectively.
3.	The experiments are comprehensive. On three commonly used benchmark datasets, the proposal achieves the best performance on both seen, unseen, and all classes. Representative SSL, open-set SSL, and NCD methods are compared. Ablation studies also show the usefulness of the proposed techniques. This clearly demonstrates the effectiveness of the proposal.

Weaknesses:
1.	The representation could be further improved. For example, there are both “unseen classes” and “unseen-classes” in the paper, this should be unified.
2.	It would be better to study the impact of the ratio of unseen classes. For example, how the performance varies with different ratios of unseen classes unlabeled examples.

---

> ### Author Response · Authors · 2022-08-02
> **Rebuttal**
>
> Thanks for your careful review and valuable comments.
>
> 1) Representation.
>
> Thanks for your valuable comments. We will uniform the style and improve the writing in the final version.
>
> 2) Different ratios of unseen classes
>
> In our experiment, we divide classes into 50% seen and 50% unseen classes and select 50% of the seen classes as the unlabeled data. Thus, in the unlabeled data set, the unseen class examples are even more than the seen classes (the ratio is 1:2), and our proposal can still achieve promising performance in this case. The performance is expected to be further improved if the number of unseen unlabeled examples decreases. And we will try to conduct experiments in a setting where all unlabeled examples are unseen classes. Thanks for your comments.

---

### Official Review · Reviewer_ZFnr · 2022-07-10

**Rating:** 8
**Confidence:** 4
**Soundness:** 3 good
**Presentation:** 3 good
**Contribution:** 3 good

**Summary:**

This paper proposes a new SSL approach, that can not only classify seen classes but also classify unseen classes automatically. Specifically, the authors assume that the unlabeled data contains classes that have not appeared in the labeled data set. This is a practical scenario and it is meaningful to study this problem. To address it, the authors propose two important techniques: pairwise similarity based unseen class classification and adaptive threshold based pseudo-label selection. Experimental results on three datasets are reported, and the results demonstrate the effectiveness of the method.

**Questions:**

1. Can the proposal be applied to other SSL methods, e.g., MixMatch, ReMixMatch, etc.
2. In some cases, the NCD methods perform worse than the baseline SSL method on unseen classes, why?
3. If we simply replace the fixed threshold in FixMatch with the new proposed threshold, could the performance be improved on standard SSL tasks?



**Ethics Review Area:**

["Responsible Research Practice (e.g., IRB, documentation, research ethics)"]

**Limitations:**

The authors addressed the limitations of their work.

**Strengths And Weaknesses:**

Strengths:
1.	The problem is novel. This paper focuses on a new problem, i.e., the unlabeled data contains unseen classes. Unlike previous robust SSL methods that discover and reject these examples, this paper tries to classify the unseen classes automatically. It is non-trivial to address this problem.
2.	The proposed method is technically sound. It is a useful technique to adopt a pairwise similarity based objective to discover new patterns. Balance the learning pace between seen and unseen classes also seems reasonable.
3.	The experimental results are convincing. Results on three benchmark datasets show the performance improvement of this proposal. Ablation studies are also convincing to show the usefulness of each technique.

Weakness:
1.	In my view, the unsupervised part is mainly based on the FixMatch algorithm. Can the proposed techniques be applied to other SSL methods?
2.	There are some typos and grammar mistakes. I suggest the author check the paper writing and improve the representation.

---

> ### Author Response · Authors · 2022-08-02
> **Rebuttal**
>
> Thank you for your review, please find the responses in the following.
>
> 1. Other SSL methods
>
> We applied the proposal to FixMatch for its simplicity and empirical success, but the proposal can still be applied to other SSL methods. The proposed UC loss and DTA loss are general techniques and can empower the existing SSL methods to classify unseen classes.
>
> 2. Performance of NCD
>
> NCD aims to classify unseen classes in the test phase, and the main idea is to transfer knowledge from labeled examples to unlabeled examples. In the semi-supervised setting, the label information is not sufficient and degenerates the performance of NCD methods.
>
> 3. Adaptive threshold with FixMatch
>
> For standard SSL tasks, the learning difficulty is the same for all classes, so we may first adopt a fixed threshold for all classes. It is expected that if the learning difficulty is different for different classes, adopting the adaptive threshold will improve the accuracy of pseudo-labels and thus improve the SSL performance.

---

### Official Review · Reviewer_jZK2 · 2022-07-11

**Rating:** 4
**Confidence:** 5
**Soundness:** 2 fair
**Presentation:** 3 good
**Contribution:** 2 fair

**Summary:**

This paper has proposed a new SSL method, Safe Unseen classification Semi-Supervised Learning (SU-SSL), which is able to classify unseen classes automatically while maintaining safeness on seen classes. Two main components are covered in this work: unseen class classification and adaptive threshold. Specifically, a novel unseen class classification objective used to exploit pairwise similarity and eliminate noisy pairs is proposed. Besides, a semi-supervised objective that adopts an adaptive threshold with distribution alignment to improve the performance on both seen and unseen classes is designed. Three commly-used datasets (i.e., CIFAR-10, CIFAR-100 and ImageNet-100) are covered in the experiments.

**Questions:**

See above.

**Ethics Review Area:**

["Inadequate Data and Algorithm Evaluation"]

**Limitations:**

Could this work be extended to other scenarios and tasks (e.g., semi-supervised object detection, etc.)?

**Strengths And Weaknesses:**

**Paper Strengths:**
>(1) This paper is easy to follow.

>(2) The structure is clear and most of the figures are easy to understand.

>(3) Some figures are interesting and intuitive.

**Paper Weaknesses:**
>(1) For one of the main contributions of this work (adaptive threshold), the novelty and motivation is not impressive. Note that the concept of adaptive threshold has been widely studied in existing SSL methods (e.g., [1], [2], etc.) and the solution of this work is relatively straightforward. Although [1] and [2] are covered in related work, it is not enough to just regard them as holistic methods without detailed descriptions. Please highlight the motivation of this contribution point. Besides, more comparison and analysis with related work should be added.

>(2) This work attempts to address unseen classes on the basis of open set semi-supervised learning. This is similar to semi-supervised few-shot learning, which contains base classes (seen classes) and novel classes (unseen classes). It is valuable to explain and analyze for this. In addition, some experiments can be added if necessary.

>(3) In addition to existing 'Analysis on Dynamic Threshold', some ablation studies and analyses about adaptive threshold can be considered to be added. For example, the advantages of dynamic threshold adjustment for specific SSL scenarios (unseen classes), comparison to other recently proposed methods using adaptive threshold, etc. Besides, It is suggested to give the specific threshold changes of each category and the visual results before and after using the adaptive threshold.

>(4) Why use Imagenet-100 instead of directly use the complete Imagenet for large-scale dataset validation experiments?

>(5) Many components are covered in Figure 2. This makes it difficult for me to master the details of the whole framework. More explanations may be helpful.

>(6) Experimental evaluation seems not sufficient. More up-to-date state-of-the-art SSL methods should be compared. Note that only ORCA is recently proposed in Table 1.

**References**

[1] B. Zhang, Y. Wang, W. Hou, H. Wu, J. Wang, M. Okumura, and T. Shinozaki, “Flexmatch: Boosting semi-supervised learning with curriculum pseudo labeling,” in NeurIPS 2021.

[2] Yi Xu, Lei Shang, Jinxing Ye, Qi Qian, Yu-Feng Li, Baigui Sun, Hao Li, Rong Jin, "Dash: Semi-Supervised Learning with Dynamic Thresholding", in ICML 2021.

---

> ### Author Response · Authors · 2022-08-02
> **Rebuttal**
>
> Thank you for your careful reviews and comments, please find the responses in the following.
>
> 1. About the contribution
>
> We should clarify that the main contribution of our paper is that we address a realistic and challenging SSL setting, i.e., the unlabeled dataset contains unseen classes. Different from the previous open-set SSL, our proposal can not only classify seen classes but also classify unseen classes. The "adaptive threshold" is only a small part of our proposal, and it is noteworthy that the "adaptive threshold" is totally different from the "dynamic threshold" in [1] and [2]. The dynamic threshold indicates the threshold changes in different iterations but all classes have the same threshold. The adaptive threshold indicates the threshold is different for different classes.
>
> 2. Semi-supervised few-shot learning
>
> The studied problem in our paper is different from semi-supervised few-shot learning. Most of the semi-supervised few-shot learning studies assume that the unlabeled data has the same class set as the labeled data. A few studies assume that the unlabeled data has unseen classes, but their goal is still to classify seen classes in the test phase. Our paper aims to classify both seen classes and unseen classes. The most related studies to our paper are open-set SSL and novel class discovery. We have discussed the relatedness and differences in Section 2 and given comparisons in the experiments.
>
> 3. Analysis of Dynamic Threshold
>
> The proposed adaptive threshold changes the threshold in different classes while the existing dynamic threshold changes the threshold in different iterations and keeps the same threshold for all classes. It is not appropriate to compare these two different techniques. The effectiveness of the adaptive threshold has been reported in Figure 5 and Table 3. The results clearly show that the adaptive threshold can improve both the number of selected examples and the accuracy of pseudo-labels.
>
> 4. Imagenet-100
>
> Imagenet-100 is widely adopted in SSL studies, such as the paper of ORCA. The data set is enough to analyze the effectiveness of different SSL methods.
>
> 5. Figure 2
>
> The main component of our proposal is the supervised loss, the UC loss, and the DTA loss. The details of UC loss and DTA loss are discussed clearly in section 4.2 and section 4.3. We will try to simplify Figure 2 to make it easier to understand. Thanks for your suggestions.
>
> 6. Compared methods
>
> For both open-set SSL and NCD, we compared with the representative methods such as the DS3L and Rankstats. The ORCA is the SOTA method in the studied problem. Compared with these methods, our proposal achieves the best performance in all commonly adopted settings. The results can demonstrate the effectiveness of our proposal clearly.

---

> > ### Comment · Reviewer_jZK2 · 2022-08-09
> > **Response to Author's Feedback**
> >
> > Thanks for the author's feedback.
> >
> > Note that the adaptive threshold proposed in FlexMatch [1] will assign different thresholds for different classes, which is not a so-called dynamic threshold. Of course, it may be designed differently. As for the specific threshold changes of each category and the visual results before and after using the adaptive threshold, I reserve my view of necessity.
> >
> > Of course, some of the responses are valuable. I'll further discuss with other reviewers and ACs.

---

> > > ### Author Response · Authors · 2022-08-09
> > > **Thanks for your response**
> > >
> > > Thanks for your response.  The difference between our paper and FlexMatch lies in the following aspects:
> > > 1) We consider a more challenging and realistic SSL scenario, i.e., the unlabeled data could contain unseen classes, while FlexMatch studies the standard SSL problem.
> > >
> > > 2)  The adaptive threshold is only a part of our proposal. We first improve propose a new unseen class classification objective that can exploit pairwise similarity and eliminate potential noisy pairs to discover unseen classes. And then adopt the adaptive threshold to
> > > bridge the performance gap between seen and unseen classes.
> > >
> > > 3) The adaptive threshold between our study and FlexMatch is also different. We propose a metric to measure the difference in the learning process between seen and unseen classes and also adopt the distribution alignment to rectify the prediction. While FlexMatch adjusts the threshold using an estimated learning effect. The mechanisms are different.

---

### Comment · Area_Chair_mGZj · 2022-08-10
**Reminder**

Dear reviewers,

Please go through the rebuttal (if you have not) and acknowledge that you have done so. Thanks!

AC

---

### Meta-Review · Area_Chair_mnWo · 2022-08-30

**Recommendation:** Accept
**Confidence:** Certain

**Metareview:**

This paper presents a method for discovering novel classes in the test data, while not deteriorating in performance on already known (seen) classes. The problem setting is similar to Novel Class Discovery (NCD) with the additional requirement that the performance on seen classes should not suffer.

The author response was discussed. In general, the paper received positive reviews. However, one of the reviewers had some concerns about the specific adaptive threshold method proposed in the work as compared to other existing adaptive threshold methods (in particular, what was the motivation behind using a new method). This aspect should be clarified in the paper.

In addition, I would like to point out that NCD with no forgetting on the seen classes has been proposed in other recent works as well, such as

Novel Class Discovery without Forgetting: https://arxiv.org/abs/2207.10659

This work should be discuss because it is solving a very similar problem.

Regardless of these concerns (which should be addressed in the final version), the paper has received largely positive reviews. Therefore I vote for acceptance.

**Award:**

No

---

### Decision · Program_Chairs · 2022-09-14

Accept